# Assessment of Agricultural Practices for Controlling *Fusarium* and Mycotoxins Contamination on Maize Grains: Exploratory Study in Maize Farms

**DOI:** 10.3390/toxins15020136

**Published:** 2023-02-07

**Authors:** Daniela Simões, Bruna Carbas, Andreia Soares, Andreia Freitas, Ana Sanches Silva, Carla Brites, Eugénia de Andrade

**Affiliations:** 1National Institute for Agricultural and Veterinary Research (INIAV), I.P., Av. Da República, Quinta do Marquês, 2780-157 Oeiras, Portugal; 2Centre for the Research and Technology of Agro-Environmental and Biological Sciences, University of Trás-os-Montes and Alto Douro (CITAB-UTAD), 5000-801 Vila Real, Portugal; 3REQUIMTE/LAQV, R. D. Manuel II, 4051-401 Porto, Portugal; 4Centre for Animal Science Studies (CECA), University of Porto, 55142 Porto, Portugal; 5University of Coimbra, Faculty of Pharmacy, Azinhaga de Santa Comba, 3000-548 Coimbra, Portugal; 6GREEN-IT Bioresources for Sustainability, ITQB NOVA, Av. da República, 2780-157 Oeiras, Portugal

**Keywords:** *Fusarium* spp., fumonisins, deoxynivalenol, fungal control, maize, mitigation practices

## Abstract

Maize is a significant crop to the global economy and a key component of food and feed, although grains and whole plants can often be contaminated with mycotoxins resulting in a general exposure of the population and animals. To investigate strategies for mycotoxins control at the grain production level, a pilot study and exploratory research were conducted in 2019 and 2020 to compare levels of mycotoxins in grains of plants treated with two fertilizers, F-BAC and Nefusoil, under real agricultural environment. The 1650 grains selected from the 33 samples were assessed for the presence of both *Fusarium* species and mycotoxins. Only fumonisins and deoxynivalenol were detected. Fumonisin B1 ranged from 0 to 2808.4 µg/Kg, and fumonisin B2 from 0 to 1041.9 µg/Kg, while deoxynivalenol variated from 0 to 465.8 µg/Kg. Nefusoil showed to be promising in regard to fumonisin control. Concerning the control of fungal contamination rate and the diversity of *Fusarium* species, no significant differences were found between the two treatments in any of the years. However, a tendency for was observed Nefusoil of lower values, probably due to the guaranteed less stressful conditions to the *Fusarium* spp. present in the soil, which do not stimulate their fumonisins production.

## 1. Introduction

Maize is one of the greatest agricultural product in human diets and is among the most important animal feed ingredient worldwide [1,2]. In Portugal, maize is a crucial crop for the national economy, having been the one with the highest production (between 8 and 15 t/ha), either of grain or silage, in recent years [3]. However, the maize crop is very susceptible to fungal infections, which can be caused by an extensive range of fungi, including mycotoxigenic species. These fungal infections are responsible for significant yield and quality losses [1], being a big concern to the farmers.

*Fusarium* is one of the major pathogenic genera affecting maize crop, causing “*Gibberella* ear rot” (associated with *F. graminearum*) in cooler temperate regions, or “*Fusarium* ear rot” (associated with *Fusarium fujikuroi* complex species) in the warmer temperate regions [4]. *Fusarium fujikuroi* complex includes *F. verticillioides*, *F. subglutinans*, and *F. proliferatum* [5], which were reported as the most frequent species found in Portuguese maize [6,7]. These species are also described as mycotoxigenic species, being the major fumonisins producers [8].

Mycotoxin contamination of raw maize or maize-based food/feed is harmful to human and animal health, resulting in acute or chronic consequences such as carcinogenic, teratogenic, immunosuppressive, or estrogenic issues [9]. For this reason, the maximum levels were set by the European Commission in raw materials, feed and processed food [10]. Additionally, contaminated maize is strongly devalued, having a direct impact in the national economy.

As mycotoxins are produced by fungi, the factors that have an impact either on the field fungi (those that invade plants before harvest) or storage fungi (those that invade grains after harvest) development can affect mycotoxin production [11]. Those factors include the substrate on which the fungus grows and environmental conditions [11]. Agricultural and post-harvest practices have clearly advanced on the way to control mycotoxins. Those practices cover controlling the fungi presence and abundance on the field, decontaminating the maize grains [12], controlling the storage conditions (temperature, humidity, ventilation, hygiene, insects’ presence, and duration) [13], and mitigating the presence of mycotoxins by adding, for instance, rice bran oil [14].

Along with the conventional monitoring of mycotoxins at the raw maize grains level, the assessment of mycotoxigenic fungal species is of upmost importance to provide useful insights, for instance, for an early control of fungi on the field, before harvesting. The overall target are prophylactic measures, i.e., more effective control practices to mitigate these fungal species’ development and proliferation. As these species are naturally present in the soil, being impossible to be eliminated, the focus will be the controlling of their behaviour towards the mitigation of mycotoxins production, since these toxins result from exposure of the fungi to stressful conditions [15]. In this way, the agricultural practices should focus on the nutritional and structural needs of both the plant and its microbiome. Several studies have shown that fertilization of the soil had positive effects on the plant tolerance to pathogenic fungi, such as using fertilizers containing either zinc or potassium phosphite [16,17,18]. The use of potassium phosphite in the growing media under greenhouse conditions resulted in lettuce protection against *Fusarium oxysporum* with a disease reduction varying from 11% to 46% [18].

The most significant mycotoxins are aflatoxins, fumonisins, ochratoxin A, deoxynivalenol, zearalenone, and ergot alkaloids which are mainly produced by the genera *Aspergillus*, *Penicillium*, *Fusarium*, and *Claviceps* [9]. In 2018, during a survey conducted in the same Portuguese region, only fumonisins B1 and B2 were detected in maize grains and were associated with the presence of *Fusarium* species [6]. These findings were the incentive for this study where the aim was to evaluate, in a real agricultural environment, the effect of two commercial fertilizers, F-BAC and Nefusoil, zinc-based and potassium phosphite-based, respectively. Both fertilizers were chosen by the farmer because they are known to improve the maize tolerance to pathogenic fungi [19,20] on both *Fusarium* species abundance and proliferation, and mycotoxins contamination. The detected mycotoxins levels were then compared with those of controls which were produced following the conventional practices.

## 2. Results

### 2.1. Fungal Contamination of Maize

The fungal analysis of 1650 (50 grains × 33 samples) putatively diseased grains collected in one farm of the Tagus valley resulted in a total of 344 *Fusarium* specimens, meaning a contamination rate of 19.5% in 2019, 22.2% in 2020 parcels, and 20.9% in 2020 random blocks.

From the 12 maize samples (four samples per treatment) collected in 2019, 117 *Fusarium* spp. isolates were detected and identified: 54 out of the F-BAC samples; 34 came from Nefusoil samples; and 29 from the control samples. Despite these different absolute numbers, the difference between treatments is not statistically significant (*p* = 0.787). The presence of *Fusarium* was very heterogenous independently of the treatment (Figure 1). Out of the four F-BAC samples, we detected 6, 5, 37 and 6 grains contaminated with *Fusarium* spp., respectively. The third sample resulted in 68.5% of all the *Fusarium* spp. isolated from F-BAC samples, what is statistically different from the other samples (*p* < 0.001), and deviated more than 1.5 times the interquartile amplitude, meaning that this value may correspond to an outlier. The Nefusoil treatment resulted in 5 to 11 contaminated grains per sample, which corresponds to 17% of maize grains contaminated with *Fusarium* spp. without significant differences between replicates (*p* = 0.365). Control samples had a contamination rate varying between 8 and 20%, averaging 14.5% of grains contaminated with *Fusarium* spp. also without significant differences between replicates (*p* = 0.460).

Out of the 117 *Fusarium* isolates, 81 (69.2%) were identified as *F. verticillioides*, 34 (29.1%) were *F. subglutinans* and only 2 (1.7%) were *F. proliferatum*, which means a predominance of *F. verticillioides* (*p* < 0.001). Under F-BAC treatment, *F. verticillioides* was the predominant species (*p* < 0.001) accounting for 82% of all the isolates. Samples either treated with Nefusoil or without treatment (control) showed an almost equal predominance of *F. verticillioides* (59%) and *F. subglutinans* (41%), without significant differences (*p* = 0.303 and *p* = 0.353, respectively).

In 2020, 133 *Fusarium* spp. isolates were detected in the samples harvested in the same parcels than in 2019: 42 originated under the F-BAC treatment; 44 under Nefusoil; and 47 in the control. The three situations were not significantly different (*p* = 0.839), with contamination rates of 21% (F-BAC), 22% (Nefusoil), and 24% (control) in average (Figure 2). Fungal contamination of maize treated with F-BAC variated between 18% and 26%, and with Nefusoil ranged from 12% to 28%. The control presented values from 16% to 32%. Species identification revealed that 86 isolates (64.7%) were *F. verticillioides*, 44 (33.1%) were *F. subglutinans* and 2 (1.5%) were *F. proliferatum*, which means that *F. verticillioides* was the predominant species (*p* < 0.001). One isolate lost its viability before the identification at the species level was finished. Therefore, it was only possible to identify it as *Fusarium* spp. Under F-BAC effect, *F. verticillioides* was the predominant species (*p* < 0.001) accounting for 79% of all the *Fusarium* isolates, while under Nefusoil or in the Control no significant differences between the frequencies of *F. verticillioides* (59% and 57%) and *F. subglutinans* (39% and 40%) were found (*p* = 0.170 and *p* = 0.238, respectively).

From the nine maize samples collected in 2020 in the random blocks, 94 *Fusarium* spp. isolates were detected and identified: 19 from F-BAC; 30 from Nefusoil; and 45 from Control, without significant differences (*p* = 0.543). In the samples treated with F-BAC the contamination rate varied between 4% and 20%, averaging 12.6% (Figure 3). Maize treated with Nefusoil had between 14% and 28% of contaminated grains which means an average of 20%. Both F-BAC and Nefusoil presented no significant intrasample variability (*p* = 0.076 and *p* = 0.273). On the contrary, control samples presented significant differences within the three replicates (*p* < 0.001): 33 (73.3%), 5 and 7 isolates were obtained out of the first, second and third replicate, respectively. Despite this observation, and as only three replicates were performed, it is not possible to calculate the interquartile amplitude, thus the first replicate was not considered as an outlier. In this way, the average of maize contaminated with *Fusarium* spp. in Control was 30%.

Out of the 94 *Fusarium* spp. isolated in the maize harvested in 2020 in the random blocks, 55 (58.5%) were identified as *F. verticillioides* and 36 (38.3%) as *F. subglutinans*. Three isolates lost their viability before their identification until the species level, so it was only possible to identify them as *Fusarium* spp. The frequency of *F. verticillioides* and *F. subglutinans* is statistically significant (*p* = 0.046), meaning a predominance of *F. verticillioides* also in the random blocks. In F-BAC and Nefusoil blocks, the incidence of *F. verticillioides* (42% and 43%) and *F. subglutinans* (47% and 53%) were not statistically significant (*p* = 0.808 and *p* = 0.577, respectively). However, in the Control, *F. verticillioides* (76%) was by far the most predominant species (*p* < 0.001).

The comparison of contamination results obtained in the parcels in 2019 and 2020, and in the random blocks is represented in Figure 4.

### 2.2. Analyses of Mycotoxins

The analyses of mycotoxins included the quantification of fumonisins B1 (Fum B1) and B2 (Fum B2), deoxynivalenol (DON), toxin T2 (T2), zearalenone (ZEA) and aflatoxins (AFB1, AFB2, AFM1 and AFM2). However, only fumonisins B1 and B2 and DON were detected. Table 1 lists these mycotoxins values for all samples in micrograms per kilogram of dry matter. All the other mycotoxins values were below the detection limits of the used methods, being considered “not detected”.

From the data set, it is readily apparent that Fum B1 was the most prominent mycotoxin in most of the samples (Table 1).

To facilitate a comparison between treatments, Figure 5 shows the average quantity and standard deviations of fumonisins B1 and B2, the only mycotoxins detected in the grains of the 2019 campaign, and Figure 6 and Figure 7 show the values of fumonisins and DON for all samples harvested in 2020 in the long plots and random blocks, respectively.

In 2019, fumonisin B1 concentration was 52% and 59% lower with F-BAC and Nefusoil than in the control, respectively. The same tendency was found for Fum B2 which was 53% lower with F-BAC and 58% lower with Nefusoil than in the control (Figure 5). The concentration of Fum B1 was always much higher than B2, in all situations except in two samples of grains grown under Nefusoil treatment in 2020 when none was detected (Figure 6).

In the campaign of 2020, either in the long plots or in the random blocks, in addition to fumonisins, deoxynivalenol (DON) was also detected (Table 1). Despite the apparent difference of mycotoxins levels between treatments, no statistical differences were found (Fum B1 *p* = 0.170; Fum B2 *p* = 0.089; DON *p* = 0.206). However, under the treatment Nefusoil, no contamination with Fum B1 was detected in two samples of grains, and Fum B2 was not detected in any of the replicates (Figure 6). In the long parcels, compared with the control, a reduction of 81% and 100%, respectively, in fumonisins B1 and B2, was showed using Nefusoil. In the other hand, using F-BAC the levels of fumonisins B1 and B2 were 32% and 44% lower than the respective controls. In the same way, in the random blocks, Nefusoil presented 85% and 100% less Fum B1 and Fum B2 than the control, while F-BAC only presented 9% less of FB1.

The concentration values of DON and Fum B2 in the grains harvested in the random blocks did not present statistical differences regardless of the treatment (*p* = 0.608 and *p* = 0.243). However, in two of the three blocks treated with Nefusoil, the goal of zero contamination of maize with Fum B2 was accomplished (Figure 7). The concentration of Fum B1 also presented significative differences between the control and Nefusoil treatment (*p* = 0.033).

## 3. Discussion

The presence of mycotoxins in the grains results in an exposure of the general population. This may happen in the occupational environments as those related with the working areas where raw materials and waste are handled (field, silos, small enterprises, etc.) or through food ingestion [21,22,23]. For example, in Europe and America where maize is used as an energy source in animal feed with an incorporation rates ranging between 50% and 80% of the diets [11], the limits established for aflatoxins and for DON were already exceeded [11,24], which is a real concern. Facing the generalized exposure to mycotoxins, it is urgent to run studies to gather information towards the development and implementation of new management strategies to reduce the presence of fungi. This is crucial considering that fungi also decrease yield [25], which means food safety and food security issues.

In this exploratory study, we gathered and analysed data on the prevalence of *Fusarium* species and mycotoxins in maize grown in a farm of the main production region of Portugal, after the use of two fertilizers to control fungal infections. To our knowledge this is the first time that a survey targets at once *Fusarium* species, their mycotoxins, and the effect of products to control the fungi survival or behaviour.

The survey of *Fusarium* species run during this study indicates that a relatively small *Fusarium* community is associated with maize grains in the farm used for the pilot and exploratory studies. The contamination rates observed in the three assays (19.5% in 2019 parcels, 22.2% in 2020 parcels, and 20.9% in 2020 random blocks) are lower than those reported in 2018 by Carbas et al. [6] (27.1%,) and Simões et al. [7] (41.6%), and mentioned in the review of Marín et al. [10] (from 25% to 100%). Additionally, the reduced contamination rates (15–30%) found in the control samples indicate that the agricultural practices adopted in this farm are beneficial. Agricultural practices may play a crucial role on the early indirect control of mycotoxins as also concluded by other authors [10,26].

Two major *Fusarium* species were recovered, as already reported in 2018 [6,7], *F. verticillioides* and *F. subglutinans*. *F. verticillioides* were the most commonly isolated species (59–65%) indicating that the climate in this region may favour its infection capability and survival, although its presence declined from 2018 to 2019 (80%) [6,7]. *F. subglutinans* accounted for 29–38% of the isolates, which represents an increase compared to the 16–18% observed in 2018 [6,7]. Some strains of *F. subglutinans* are described as capable of producing fumonisins, but in lower amounts than *F. verticillioides* [27], which is the major producer of fumonisins. Therefore, the reduction in *F. verticillioides* presence made us also expect a reduction in the fumonisins concentration, which is what happened. In this way, the fluctuation between *F. verticillioides* and *F. subglutinans* incidences may be correlated to different composition of mycotoxins between the two studies (Figure 5 and Figure 6). This fluctuation may, in turn, be due to factors such as climate effects (as the increase in the rainfall observed in 2020) [11,28], sample size, target organ, spatial distribution of the pathogens, and the plants microbiome. The experimental design adopted in 2020 allowed us to exclude the factors sample size (all samples were of the same size), target organ (all samples were composed of seeds) and spatial distribution (the species distribution in 2020 was identical either in the long plots or in the random blocks). However, co-presence and facilitation of subsequent infections are known in the *Fusarium* genus [29], which may mask the presence of species with lower representativity or competitive capability. Additionally, infection with multiple *Fusarium* species has been shown to affect disease severity and mycotoxin contamination [30]. Unfortunately, we still did not implement the mechanism to undoubtedly determine multiple infections (metabarcoding methods) and predict disease development and mycotoxin production. Further studies on the microbiome structure linked with the knowledge of annual variation in *Fusarium* predominant species will determine if the climatic conditions together with the plant microbiome structure are the cause for the proliferation of some *Fusarium* species. That will allow to anticipate decisions regarding the direction of crop management, disease control and mycotoxins mitigation.

All the mycotoxins were at levels below the maximum limits established by the EU [31]. The samples were contaminated mostly with three different mycotoxins, fumonisins B1 and B2, and DON, being the detection of DON (only in 2020) the first detection in Portuguese maize grains [6,32]. Despite maize be a crop with possible contamination with high levels of also aflatoxins, OTA, and ZEA, these marked geographic variations are expected [11], so their non-detectable values were not surprizing. Furthermore, the data gathered on the *Fusarium* species composition are in line with the mycotoxins detected. The presence of few toxigenic fungi species, as we observed, may be the reason for not having a multiple toxin contamination. Fumonisins B1 and B2 were present in all plots but not affecting all samples. A rough comparison between the levels found in all controls and in 2018 [6] shows that there was a tendency to decrease the concentration of both fumonisins. This decrease may be associated with a complex interaction among several factors including the environmental conditions, insect infestation, pre and post-harvest handling [25], and predominance of fungi species [28]. In 2020, fumonisins B1 and B2 decreased, and DON was for the first time detected. Curiously, DON is a toxin produced mainly by *Fusarium graminearum* and *Fusarium culmorum* [33] and is often associate with wheat rather than with maize [34]. Even though neither *F. graminearum* nor *F. culmorum* have been isolated. However, *F. verticillioides* is known as a successor of *F. graminearum* with better survival capability [29], which can explain the detection of *F. verticillioides* but not of *F. graminearum*. Some authors referred that direct sowing may significantly affects plants’ health status, expressed by an increase in mycotoxin accumulation [19]. This means that DON occurrences could be explained also by the direct sowing of maize used in 2020. Despite DON has been found in breakfast cereals in Portugal in the range of 46–525 µg/Kg [35] and with an average residue level of 5.1 µg/Kg [19], these occurrences were by far originated in wheat rather than in maize [19]. However, there are some reports of DON as a relevant contaminant of poultry and pig feed, in other European and American countries where *F. graminearum* and *F. culmorum*. are the major field cereal pathogens [11,36]. Although there are several agricultural measures and methods to control mycotoxins (physicochemical, biochemical, molecular, and biocontrol), they are not used in contaminated maize [34]. In this study, we gathered data from both a pilot and an exploratory study evaluating the putative effectivity of two fertilizers (F-BAC and Nefusoil) to control the plant performance against natural occurrence of *Fusarium* species in maize grains. The great advantages of these products are that they do not contaminate the environment, do not leave residues neither in plants nor in grains, and do not need special technology to be applied. Apparently, both products did not influence the diversity and frequency of the *Fusarium* species. The contamination rate of grains originated in plants treated with Nefusoil was 17–22%, with F-BAC was 13–27%, and from the control was 15–30%. However, the effectiveness in reducing fumonisin contamination was observed, i.e., in the presence of Nefusoil the contamination with FB2 was reduced to zero. The use of Nefusoil resulted in a more evident reduction in fumonisins (Table 1), varying between 58% and 100%. In 2020, fum B2 was not detected after the treatment and Fum B1 decreased 81–85% when comparing to the respective controls. However, F-BAC did not provide so evident benefits on the control of fumonisins. Its effect was more variable ranging the reduction between 0% (no effect) and 53%.

The positive effect of nitrogen/phosphor fertilizer at the sowing time was already described [19], but nothing was so far reported concerning the effect of potassium (Nefusoil) or zinc (F-BAC) in the *Fusarium* species diversity and mycotoxin contamination of maize produced in real conditions. A possible explanation for the favourable effects of Nefusoil lies in its action on soil-born microorganisms rather than on the plant performance. Nefusoil stimulates the action of saprophytic microorganisms in the soil, and the synthesis of secondary plant metabolites [19]. *Fusarium* species are saprophyte, naturally present in the soil, which in the presence of susceptible hosts can cause infection, and under stressful conditions can produce mycotoxins [15]. As no significative differences of contamination rates with *Fusarium* spp. were found in grains between Nefusoil and the control, the use of Nefusoil did not apparently inhibit the entrance of *Fusarium* into the roots and their progression until the grains. Thus, the main effect of Nefusoil is not reflected in the plant but is reflected in the fungal behaviour. It appears to guarantee less stressful environmental conditions to the *Fusarium* species present in the grains, which does not stimulate them to synthesize mycotoxins, and consequently not increasing the mycotoxin concentration. The number of contaminated grains with fungi does not *per se* explain the occurrence/absence of mycotoxins.

## 4. Conclusions

The results of these pilot and exploratory studies extended considerably our knowledge on the effect of these two fertilizers on the control of *Fusarium* mycotoxins in maize fields. Nefusoil presented the best results in reducing fumonisins presence in maize, showing a reduction of up to 85% of fumonisin B1 and 100% of fumonisin B2. Nefusoil together with the agricultural practices adopted by this farmer appear to be beneficial to control the fungal behaviour and fumonisins presence. It appears to guarantee less stressful conditions to fungi, preventing their stimulation to produce fumonisins. The common practices used by this farmer appear to be beneficial, except the direct sowing, which appears to be related to the increase in DON contamination.

Our survey updated on the status of fumonisins B1 and B2, and DON in maize produced in Portugal. The concentration of all detected mycotoxins remained below the limits established by the European Commission and lower than in the previous studies which allows us to state that our maize is safe.

## 5. Materials and Methods

### 5.1. Maize Cultivation Conditions

The studies were conducted during the maize-growing seasons of 2019 and 2020, in one farm in the Tagus Valley region, in Portugal.

In 2019, a pilot study following the farmer initiative was conducted in 3 parcels (two treatments plus the control) with approximately 15,050 m^2^ each, embracing 48 lines at the density of 90,000 seeds/ha (Figure 8, longitudinal rows). Two treatments were tested to understand the potential effect on plant protection: (1) F-BAC (EIBOL Ibérica S. L., Valencia, Spain), a fertilizer with nitrogen, phosphor, and zinc (N:P_2_O_5_:Zn, 5:15:1.5) that stimulates the plant metabolism [20]; and (2) Nefusoil (EIBOL Ibérica S. L. Valencia, Spain), a fertilizer composed by nitrogen, phosphor, and potassium (N:P_2_O_5_:K_2_O, 4:12:3) that stimulates the saprophyte microorganisms action on the soil [19]. Both fertilizers were used at the minimal recommended doses, 1.5 L/ha and 15 L/ha, respectively. The study run under real farm conditions, with the current agricultural practices (standard fertilizer DAP 18-46-0 applied at approximately 125 Kg/ha), weather conditions, and soil-type. In 2020, we run an exploratory study to validate the results of the pilot study and to confirm the putative interest of the tested products. Therefore, the same parcels were set in 2019 to estimate repeatability, and additionally, a new assay was designed in random blocks with three replicates for each treatment (Figure 8, transversal rows), totalizing around 9900 m^2^ per treatment. Only one cultivar has been used (Pioneer P0933) due to the preferences of the farmers of that region.

In Portugal, maize is a Spring–Summer crop. The sowing dates varied according to the weather conditions. In 2019, sowing occurred within the two first weeks of March and the harvesting within the two first weeks of October. In 2020, due to intensive rain occurrence in March, sowing was performed in the fourth week of March, under minimal soil disturbance.

Between March and October 2019, the maximal temperature was 32 °C (in August) and the minimal temperature was 8.1 °C (in March) (Figure 9), with the average temperature varying between 14.9 °C and 24.2 °C. In the same period of 2020, the maximal temperature was 35.7 °C (in July) and the minimal temperature was 8.8 °C (in March) (Figure 9), with the average temperature varying between 14.2 °C and 26.3 °C. During these seven months, in 2019, the rainfall varied between 0.9 mm (in July) and 92.2 mm (in April), with an average of 25.9 mm, while in 2020, the rainfall varied between 0.9 mm (in August) and 121.5 mm (in April), with an average of 38.7 mm (Figure 9). In 2020, the rainfall was higher in April and in October at the harvesting season.

### 5.2. Crop Harvest and Sampling

The long plots (FB, Nef, and Control—Figure 8), each with 48 lines of maize plants per treatment, were divided in four sectors (each with 12 lines), corresponding approximately to 3762 m^2^ each. The grains harvested in each sector of 12 lines got mixed in the harvester hopper and were transferred to the trailer of the tractor, where sampling occurred. Sampling has been performed using a vertical segmented probe with 7 chambers, which allowed for the sampling of maize along the entire depth. Each sample consisted of 10 sub-samples, which means that 70 increments were taken from the top to the bottom of the trailer (approx. 5 Kg). In this way, 20 Kg of each treatment (280 increments) were collected, ensuring the representativity of the entire field. At the maturity stage, grains had 17–19% moisture.

Each random block also had 12 lines (with approximately 3300 m^2^), and the sampling followed the same procedure described above, totalizing 3 independent samples per treatment (approx. 5 Kg each, i.e., around 15 Kg per modality).

In total, 33 samples of 5 Kg of maize grains were collected: 12 from long parcels in 2019; 12 from long parcels in 2020; and 9 from random blocks in 2020.

### 5.3. Fungal Detection, Identification, and Quantification

Climate is an important factor that modify fungal development and leads to variable levels of mycotoxins [11]. In temperate regions, colonization of maize by *Fusarium* species and consequent contamination with fumonisins are expected [11]. In this way, and following our previous studies [6,7], the mycological analyses were focused on *Fusarium* spp. detection and identification. The preparation of samples was performed as described by Carbas and co-workers [6]. Briefly, 50 grains per sample were surface disinfected and plated aseptically on Malachite Green Agar 2.5 [37] at the rate of five grains per 90 mm Petri-dish: one in the centre and one in each quadrant. Plates were incubated at 25 °C for 12 h under near UV light and 12 h darkness [38]. After 7 days, the quantification of maize grains contaminated with *Fusarium* was performed, and the *Fusarium* specimens were isolated by the single spore technique [5] onto Potato Dextrose Agar (PDA), BD Difco™, (Fisher Scientific, Porto Salvo, Portugal) and incubated for 7 days. Fungal culture from each diseased grain represents an independent isolate. The isolates were grouped by morphotypes, picked to Carnation Leaf-piece Agar (CLA) and Spezieller Nährstoffarmer Agar (SNA) media, and after 10 days of incubation, were identified until species level by macro and microscopic (100× and 400×) morphology, following Leslie and Summerell (2006) [5].

### 5.4. DNA Extraction and Molecular Identification

Either to complete or to confirm the species identity of the isolated *Fusarium* pathogens, the recommendation of Carbone and Kohn [39] was followed. The sequence of the variable fragment of the *Translation Elongation Factor 1-α* (*TEF*) gene was obtained after amplification with primers EF1-728F and EF1-986R. This procedure was applied in all the cases where the species of individual isolates could not be identified with certainty based only on their colony morphology. Additionally, it was also used to validate the protocol for morphological identification, randomly selecting and then sequencing about 10% of all the isolates. In total, 79 isolates were sequenced.

Total genomic DNA was extracted from each isolate using Chelex^®^ 100 Chelating Resin, analytical grade, 200–400 mesh, (Bio-Rad Laboratories, Richmond, CA, USA) [40,41]. *TEF* gene amplification and sequencing was performed following the method described in Carbas et al. (2021) [6]. After editing the nucleotide sequences, using the Chromas Lite software, Technelysium (South Brisbane, Australia), the sequences were compared with the database NCBI (https://blast.ncbi.nlm.nih.gov/Blast.cgi (accessed on 1 April 2021)) and only accepted when both the coverage and homology were equal or higher than 98%.

### 5.5. Mycotoxins’ Extraction

In the pilot study of 2019, mycotoxins were analysed in one composite sample per treatment. This should be taken simply as a screening procedure. In 2020, the mycotoxins analyses were performed separately in each of the four samples harvested per treatment just the fungal analyses were performed. To extract the mycotoxins, the analytical method described by Silva et al. was used [42]. Maize grains were thoroughly mixed and approximately 1 Kg was ground in a Retsch rotor mill (SK300) with a sieve of trapezoid holes of 1.00 mm. Maize flour was stored at −20 °C until further analysis. Maize flour was extracted with acetonitrile 80% (*v/v*) in an orbital shaker (Kotterman 4010, Uetze/Hanigsen, Germany), for 1 h. To analyse fumonisins, the centrifugated extract was diluted with ultra-pure water. For the other mycotoxins, the centrifugated extract was evaporated by gentle stream of nitrogen at 40 °C and the residue was redissolved with acetonitrile 40% (*v/v*).

### 5.6. Mycotoxin UHPLC-ToF-MS Analyses

The detection and quantification of fumonisins (B1 and B2), toxin T2 (T2), zearalenone (ZEA) and aflatoxins (AFB1, AFB2, AFM1 and AFM2) were performed according to the method described and validated by Silva et al. [42], using a Nexera X2 Shimadzu Ultra-High Performance Liquid Chromatography (UHPLC) coupled with a 5600 Time-of-Flight Mass Spectrometry (ToF-MS) detector (SCIEX, Foster City, CA, USA) equipped with a Turbo Ion Spray electrospray ionization source working in positive mode (ESI). For separation, a column Zorbax Eclipse Plus C18 (2.150 mm, 1.8 μm) was used. In the acquisition by mass spectrometry, the Analyst^®^TF (SCIEX, Foster City, CA, USA) software was used in full scan from 100 to 750 Da. The data processing was performed using PeakView™ and MultiQuant™ (SCIEX, Foster City, CA, USA) software. The quantification limits (LoQ) previously reported [42] are for AFB1 (1 μg/Kg), AFB2 (1 μg/Kg), AFG1 (1 μg/Kg), OTA, (1,5 μg/Kg), Fum B1 (125 μg/Kg), Fum B2 (125 μg/Kg), T2 (25 μg/Kg) and ZEA (50 μg/Kg). The analytical technique has sensitivity enough to meet the requirement imposed by European Union (EU) regulations for the maximum limits of mycotoxins in maize, except for baby food.

### 5.7. Deoxynivalenol (DON) Analyses

The detection and semi-quantitative screening of deoxynivalenol (DON) in maize samples were performed using a chemiluminescent biochip immunoassay based on the Evidence Investigator Biochip Array technology (Investigador™ EV 4065), previously validated by Freitas et al. [43]. For the extraction procedure of DON, 5 g of homogenized sample were extracted with 25 mL acetonitrile:methanol:water (50:40:10, *v/v/v*). After, they were vortexed (60 s), rolled (10 min) and centrifuged at 3000 rpm for 10 min. Then, they were diluted with working strength wash buffer (included in the kit), 50 μL sample and 700 μL working strength (dilution factor: 75). The diluted sample was applied to the biochip of the assay Myco 7, according to the instructions of the manufacturer (Randox Laboratories Limited 2008, Crumlin, UK), which contains regions with immobilized antibodies restricted to mycotoxins.

### 5.8. Statistical Analyses

Statistical analyses were performed using the software IBM SPSS Statistics 28.0.0.0 (IBM Corp, Armonk, NY, USA). X^2^ was preformed to analyse the significance of proportions. Levene’s test and Shapiro–Wilk test were performed to analyse the homogeneity and normality of the variable distribution, and then, whenever possible the ANOVA test was performed, otherwise, the Kruskal–Wallis’ test was applied. The results were considered significantly different only if *p* < 0.05 (α = 0.05).

## Figures and Tables

**Figure 1 toxins-15-00136-f001:**
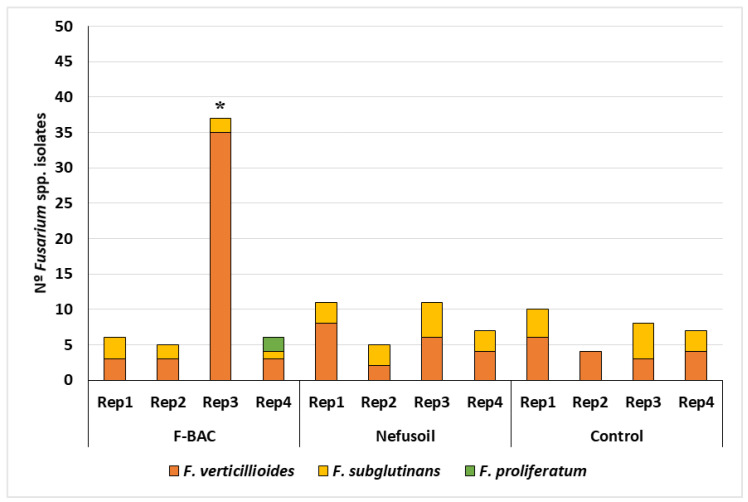
Results of maize contamination with *Fusarium* spp. and respective distribution of species, per sample and treatment, in 2019. * Corresponds to an outlier.

**Figure 2 toxins-15-00136-f002:**
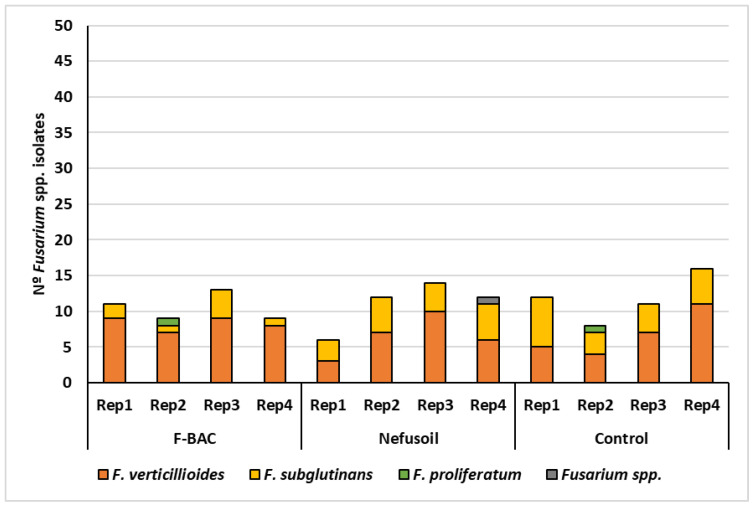
Results of maize contamination with *Fusarium* spp. and respective distribution of species, per sample and treatment, harvested in 2020 in the same parcels than in 2019.

**Figure 3 toxins-15-00136-f003:**
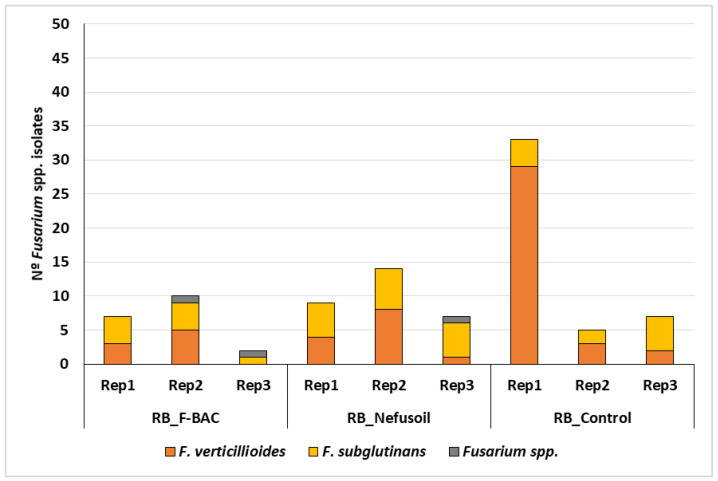
Results of maize contamination with *Fusarium* spp. and respective distribution of species per sample and treatment, harvested in 2020 in the random blocks.

**Figure 4 toxins-15-00136-f004:**
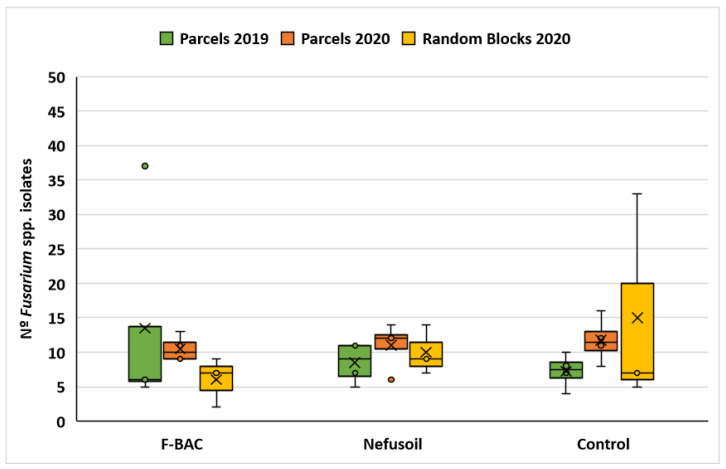
Comparison of maize contamination with *Fusarium* spp. results, per year, parcels (of 2019 and 2020) and random blocks (2020), and treatment (F-BAC, Nefusoil and Control). The coloured box represents the values between the 1st and the 3rd quartiles, the bars represent the maximum and the minimum values, the cross represents the average and the squares represent outlier points.

**Figure 5 toxins-15-00136-f005:**
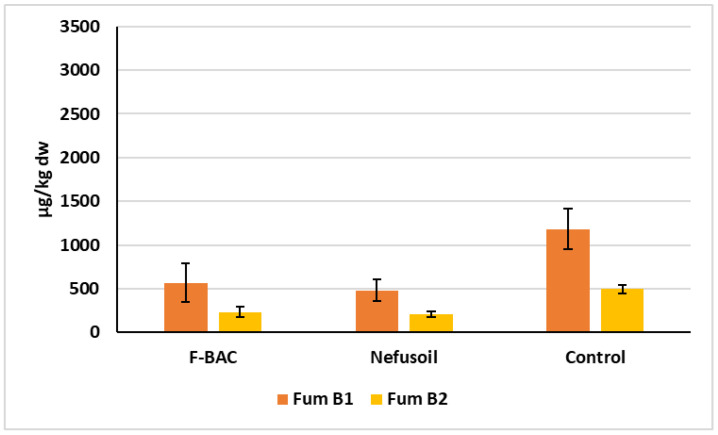
Levels of mycotoxins detected on the maize samples harvested in 2019, from the plots treated with F-BAC and Nefusoil, and the plot untreated (Control).

**Figure 6 toxins-15-00136-f006:**
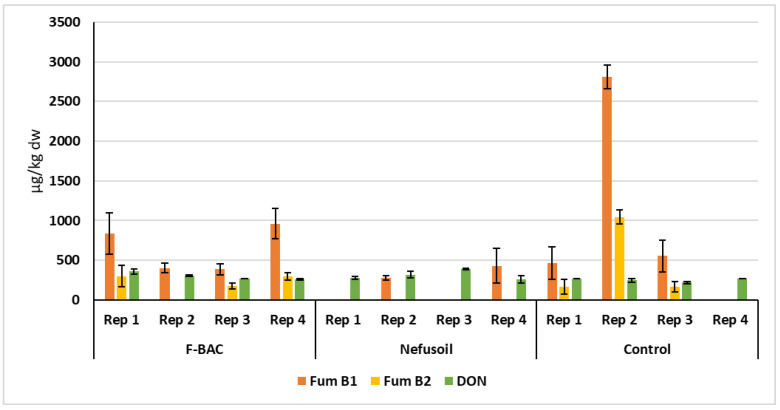
Levels of mycotoxins detected on maize samples harvested in 2020 from the parcels (longitudinal rows of Figure 1) treated with F-BAC and Nefusoil, and from the Control plot.

**Figure 7 toxins-15-00136-f007:**
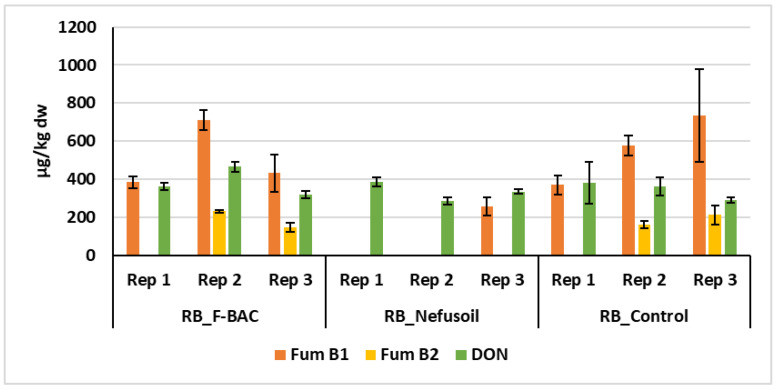
Levels of mycotoxins on maize samples harvested in 2020 in the random blocks treated with F-BAC (RB_F-BAC) and Nefusoil (RB_Nefusoil), and untreated (RB_Control).

**Figure 8 toxins-15-00136-f008:**
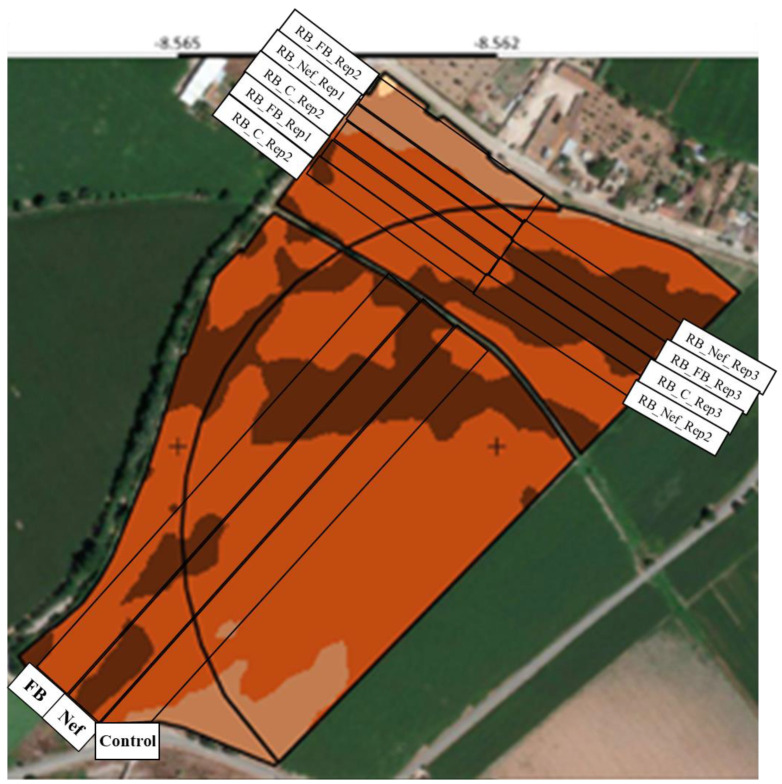
Design of the studies to evaluate the effect of treatments with F-BAC (FB) and Nefusoil (Nef), and the Control (C) in parcels in 2019 and 2020 (longitudinal rows) and random blocks (RB) in 2020 (transversal rows).

**Figure 9 toxins-15-00136-f009:**
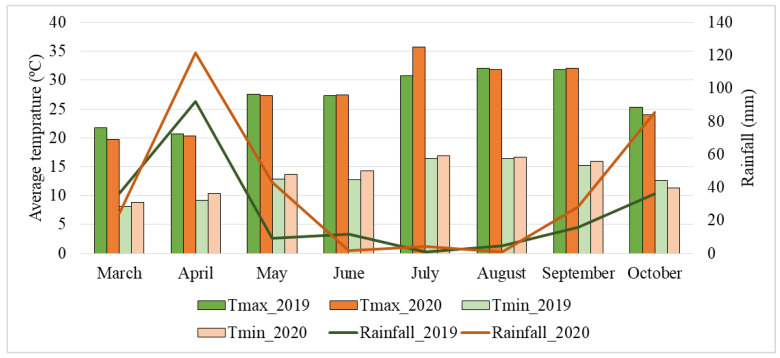
Total rainfall (mm), and maximal and minimal temperatures (°C) registered in 2019 and 2020 between March and October. (Assessed at INE, IP: https://www.ine.pt/) (accessed on 10 March 2021).

**Table 1 toxins-15-00136-t001:** Mycotoxin levels in all samples (long parcels of 2019 and 2020; and random blocks of 2020). * Difference in percentage between the levels of Fum B1 and Fum B2 detected in the long parcels in 2019 and 2020. n.d—not detectable.

Treatment	Sample (Year)	Fum B1 Mean (µg/Kg dw)	Variation between Years * (%)	Fum B2 Mean (µg/Kg dw)	Variation between Years * (%)	Deoxynivalenol Mean (µg/Kg dw)
**F-BAC**						
	Long parcel 2019	568.8	12.0	231.5	16.6	n.d.
	Long parcel 2020	646.1		193.1		298.4
	Random blocks	509.6		125.2		382.5
**Nefusoil**						
	Long parcel 2019	480.8	63.1	207.6	100.0	n.d
	Long parcel 2020	177.2		n.d		309.8
	Random blocks	85.3		n.d		335.3
**Control**						
	Long parcel 2019	1182.4	19.1	495.7	30.6	n.d
	Long parcel 2020	956.4		343.9		251.6
	Random blocks	560.5		124.0		343.8
Years before						
	Survey 2018 [6]	1666.75		922.7		

## Data Availability

Not applicable.

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
