# Peer review of "Assessment of Agricultural Practices for Controlling Fusarium and Mycotoxins Contamination on Maize Grains: Exploratory Study in Maize Farms"

_toxins, 2023, doi:10.3390/toxins15020136_

Round 1

Reviewer 1 Report

The article, Assessment of agricultural practices for controlling Fusarium and Mycotoxins Contamination on Maize Grains, presented for review, concerns a very important issue of the occurrence and prevention of the growth of microscopic fungi and their metabolites on a raw material of significant economic importance.

Combating this type of contamination already at the stage of raw material production is of key importance for the quality and safety of products obtained from them. It was therefore with great interest that I read the manuscript submitted for review.

I am unable to evaluate self-citations due to the hidden names of the Authors.

The study is written in the correct language, both the introduction, the purpose and the description of the methods do not raise any doubts.

During the revision, several inaccuracies were noticed, hence the request for correction or additional explanations:

1. Please standardize the size and type of fonts in drawings. There is a lot of chaos in it. Some drawings have a TNR font and others do not, such as Figures 4 and 5.

2. Please explain why DON was not determined by reference chromatographic methods.

3. In the methodological part, the authors write "The detection and quantification of fumonisins (B1 and B2), toxin T2 (T2), zearalenone (ZEA) and aflatoxins (AFB1, AFB2, AFM1 and AFM2) were performed according..." I did not find the results of T2, ZEA and aflatoxin determinations. What happened to these results, were the amounts below the detection limit? Honestly, I would rather expect a large amount of zearalenone itself.

Please unify the text in this regard by supplementing the data or updating the methodological part.

4. Please supplement the literature review or discussion with studies on the occurrence of mycotoxins in feed, e.g. Cegielska-Radziejewska R., Stuper-Szablewska K., Szablewski T. 2013. Microflora and mycotoxin contamination in poultry feed mixtures from western Poland. Annals of Agricultural and Environmental Medicine. 20(1): 30-35.

Author Response

Dear reviewer

Many thanks for helping us with this Manuscript. On behalf of the team involved in these studies, I am very grateful with your suggestions that were very helpful to improve our article. The corrections were incorporated in the revised manuscript. The following text has detailed point-to-point responses.

“The article, Assessment of agricultural practices for controlling Fusarium and Mycotoxins Contamination on Maize Grains, presented for review, concerns a very important issue of the occurrence and prevention of the growth of microscopic fungi and their metabolites on a raw material of significant economic importance. Combating this type of contamination already at the stage of raw material production is of key importance for the quality and safety of products obtained from them. It was therefore with great interest that I read the manuscript submitted for review. I am unable to evaluate self-citations due to the hidden names of the Authors. The study is written in the correct language, both the introduction, the purpose and the description of the methods do not raise any doubts.”

During the revision, several inaccuracies were noticed, hence the request for correction or additional explanations:

  1. Please standardize the size and type of fonts in drawings. There is a lot of chaos in it. Some drawings have a TNR font and others do not, such as Figures 4 and 5.

Done.

  1. Please explain why DON was not determined by reference chromatographic methods.

The UHPLC-ToF-MS method adopted was not validated for DON determination due to the operational constrains.

  1. In the methodological part, the authors write "The detection and quantification of fumonisins (B1 and B2), toxin T2 (T2), zearalenone (ZEA) and aflatoxins (AFB1, AFB2, AFM1 and AFM2) were performed according..." I did not find the results of T2, ZEA and aflatoxin determinations. What happened to these results, were the amounts below the detection limit? Honestly, I would rather expect a large amount of zearalenone itself. Please unify the text in this regard by supplementing the data or updating the methodological part.

All the samples presented values below the detection limits to T2, ZEA and aflatoxins. This clarification was added to the section 2.2 and 5.6.

  1. Please supplement the literature review or discussion with studies on the occurrence of mycotoxins in feed, e.g. Cegielska-Radziejewska R., Stuper-Szablewska K., Szablewski T. 2013. Microflora and mycotoxin contamination in poultry feed mixtures from western Poland. Annals of Agricultural and Environmental Medicine. 20(1): 30-35.

We accepted the recommendation and introduced in line 314 of the revised version.

Reviewer 2 Report

Immediately after reading the article, I suggest recommending it to the authors or publishers to forward it to another journal for review, namely Agronomy (MDPI). I sincerely believe that this article will be appropriate there. I wish the authors good luck and I want to offer them some recommendations on their article:

I propose to define directly in the Abstract what F-BAC and Nefusoil are. It follows from the key distribution that Nefusoil is a fertilizer, on page 2 (lines 66-67) it is indicated that these are two commercial products (known as inhibitors of fungi proliferation on maize), and what F-BAC is remains unclear until page 8 (line 256), where it is written that this is also a fertilizer. And moreover, it is written that Nefusoil is potassium, and F-BAC is zinc (lines 270-271)? How can they inhibit the development of fungi? Is the mechanism known? Since we are talking about publication in some scientific journal, so I would like to ask authors to add these explanations. At the same time, the authors themselves write (lines 272-275) that "Nefusoil stimulates the action of saprophytic microorganisms in the soil, and the synthesis of secondary plant metabolites [27]. Fusarium species are saprophyte, naturally present in the soil", which means that there is and cannot be any inhibition of fungal growth, but there is growth stimulation. In the article there are obvious contradictions between the initial presentation of the material and the conclusions about the action of microelements like potassium and zinc (what are the anions of the salts?).

I suggest that the authors characterize the compositions of these commercial drugs, since it is obvious that there are some mixtures of active substances there, and give this information about two commercial drugs. It is necessary to present the motivation in the Introduction, why exactly these 2 substances were chosen for analysis in this article, since it already follows from the data in Figure 1 that the drugs do not reduce the level of damage to raw maize mushrooms in any way in comparison with the control, and in some cases this lesion exceeds the control.

Please indicate what doses of substances were introduced. Without this, these data and their discussion become incomplete: maybe the concentrations used were not effective, or maybe the drugs themselves do not work as effectively as expected. And what was expected from these drugs based on the results of their in vitro tests?

Figure 3: In the caption, give the abbreviation (RB) after the words “random blocks".

Author Response

Dear reviewer

Many thanks for helping us with this Manuscript. On behalf of the team involved in these studies, I am very grateful with your suggestions that were very helpful to improve our article. All the suggestions but the journal chosen for the publication were accepted. Corrections were incorporated in the revised manuscript. The following text has detailed point-to-point responses.

Immediately after reading the article, I suggest recommending it to the authors or publishers to forward it to another journal for review, namely Agronomy (MDPI). I sincerely believe that this article will be appropriate there. I wish the authors good luck and I want to offer them some recommendations on their article:

This is a multidisciplinary study involving agriculture, chemistry, mycology among others disciplines. Indeed, we agree that the subject could be published in different journals. We opted by toxins as the final goal is to reduce mycotoxins in maize-based food and feed.

I propose to define directly in the Abstract what F-BAC and Nefusoil are. It follows from the key distribution that Nefusoil is a fertilizer, on page 2 (lines 66-67) it is indicated that these are two commercial products (known as inhibitors of fungi proliferation on maize), and what F-BAC is remains unclear until page 8 (line 256), where it is written that this is also a fertilizer.

We introduced alterations in the abstract, “key contribution” and section 5.1..

And moreover, it is written that Nefusoil is potassium, and F-BAC is zinc (lines 270-271)? How can they inhibit the development of fungi? Is the mechanism known? Since we are talking about publication in some scientific journal, so I would like to ask authors to add these explanations. At the same time, the authors themselves write (lines 272-275) that "Nefusoil stimulates the action of saprophytic microorganisms in the soil, and the synthesis of secondary plant metabolites [27]. Fusarium species are saprophyte, naturally present in the soil", which means that there is and cannot be any inhibition of fungal growth, but there is growth stimulation. In the article there are obvious contradictions between the initial presentation of the material and the conclusions about the action of microelements like potassium and zinc (what are the anions of the salts?).

Being our work an exploratory study, we did not intend to study the mechanisms, but rather to evaluate the result of the application of these fertilizers in real agricultural environment.

I suggest that the authors characterize the compositions of these commercial drugs, since it is obvious that there are some mixtures of active substances there, and give this information about two commercial drugs.

We added the full composition of the two fertilizers as indicated in the website of Eibol in section 5.1.:

F-BAC = (N:P2O5:Zn, 5:15:1.5)

Nefusoil = (N: P2O5:K2O, 4:12:3)

It is necessary to present the motivation in the Introduction, why exactly these 2 substances were chosen for analysis in this article, since it already follows from the data in Figure 1 that the drugs do not reduce the level of damage to raw maize mushrooms in any way in comparison with the control, and in some cases this lesion exceeds the control.

We appreciate your recommendation to refer the motivation in the introduction. However, we mentioned in material and methods that the two fertilizers were studied because they were of the preference of the farmer.

We added to the last paragraph of the introduction (lines 81-82 of the revised version) this explanation.

Please indicate what doses of substances were introduced. Without this, these data and their discussion become incomplete: maybe the concentrations used were not effective, or maybe the drugs themselves do not work as effectively as expected. And what was expected from these drugs based on the results of their in vitro tests?

The requested information was added to the point 5.1 of material and methods

Figure 3: In the caption, give the abbreviation (RB) after the words “random blocks".

Done

Author Response

Dear reviewer

Many thanks for helping us with this Manuscript. On behalf of the team involved in these studies, I am very grateful with your suggestions that were very helpful to improve our article. The corrections were incorporated in the revised manuscript. The following text has detailed point-to-point responses.

The present study entitled “Assessment of agricultural practices for controlling Fusarium and Mycotoxins Contamination on Maize Grains: exploratory study in maize farms” analyzed the presence of Fusarium species and the accumulation of mycotoxins in maize crops and maize-based products during 2019 and 2020, and the use of two commercial products as a mitigation strategy. The topic and aim have been well-introduced, a comprehensive sampling has been carried out and results have been thoroughly discussed. Nevertheless, there are some issues that need to be addressed.

  • A brief explanation about the two assayed compounds must be included within the introduction section.

Done.

  • The samples used for the quantification of mycotoxins are composite maize samples? Maize flour? In that case, authors need to take into consideration that the processing of raw materials, such as maize grains, could affect the total burden of mycotoxins.

In 2019, one composite sample was prepared per treatment, but in 2020 four samples were analysed separately (lines 386-388). In both years, maize grains are ground to obtain flours.

The following sentence was included:

The samples were thoroughly mixed and approximately 1 kg was ground in a Retsch rotor mill (SK 300) with a sieve of trapezoid holes of 1.00 mm and stored at −20 °C for further analysis.

  • The presence of aflatoxins in maize is a well-documented fact. Nevertheless, I am curious about why authors decided to evaluate the presence of aflatoxins but the fungal analysis only included Fusarium species, since aflatoxins are produced by Aspergillus species.

From other studies that we conducted on the same field, we expected that fumonisins were the main mycotoxins. Actually, this was also what we observed in this study. However, for precision, all the samples were analyzed to determine the T2, ZEA, aflatoxins, DON, and fumonisins. As only DON and fumonisins were detected, so Fusarium was the select genus to be studied.

The selection of mycotoxins for the analysis was based on the published list in European Regulation (EC) No 118/2006 and aflatoxins was not detected. The fungal analysis was determined after mycotoxin results so, focused on the fumonisin and DON fungi producer’s species.

  • In lines 238-239 authors mentioned: “The presence of few toxigenic fungi species, as we observed, may be the reason for not having a multiple toxin contamination”. Sometimes, a sample can test as false negative if the analytical technique is not sensitivity enough. Could authors provide the limits of quantification of the analytical technique for the assessment of mycotoxins?

The quantification limits (LoQ) previously reported (Silva et al, 2019) are for AFB1 (1 μg/kg), AFB2 (1 μg/kg), AFG1 (1 μg/kg), OTA, (1,5 μg/kg), FB1 (125 μg/kg), FB2 (125 μg/kg), T2 (25 μg/kg) and ZEA (50 μg/kg). The analytical technique has sensitivity enough to meet the requirement imposed by EU regulations for the maximum limits of mycotoxins in maize, except for babyfood. This was added in Material and Methods.

  • Lines 251-252. Are authors sure that in-text reference [27] corresponds to the one indicated in the reference list at the end of the manuscript? I could not access the link. Please check the references list.

That is correct. The link is https://www.eibol.com/pt-pt/productos/nefusoil-regenerador-suelo-mantiene-equilibrio-biologico/ and corresponds to the web site about the fertilizer “Nefusoil”.

Reviewer 4 Report

The present paper assessed the potential of agricultural practices in the control of Fusarium and mycotoxins contamination in maize grains. The findings of the paper is of interest especially to maize farmers and other stakeholders including consumers.

Major comments

The authors need to thoroughly work on the manuscript to correct language flaws.

Comments

Line 12, change “the mycotoxins” to “mycotoxin”

Line 12, first time use of the abbreviation “DON”. Write the meaning

Line 16-18, correct sentence

Line 20-23, correct sentence

Line 31-32, correct sentence

Line 107-109, correct sentence

Line 113, “de”?

Line 184-186, Is this supposed to be part of the text?

Line 191-194, correct sentence

Line 217 – 222, the sentence is too long and confusing. Restructure and correct the sentence.

Line 218 -219, “attributed to differences in species composition”. How?

Line 253-255, what about the physical control methods?

Line 262-264, “However, some effectiveness…” Difficult to understand what the authors mean

Line 266-268, “However, F-BAC did not provide benefits…” This statement contradicts your earlier statement.

Line 293 – 294, correct sentence

Line 297, stablished?

Line 396, the manuscript for the first time mentioned that other mycotoxins including zearalenone and aflatoxins were analysed. This was not reported in the result and discussion sections. Even though the samples were negative of these toxins. The authors should report this in the result and discussion with possible explanation why this trend was observed in the study.

Author Response

Dear reviewer

Many thanks for helping us with this Manuscript. On behalf of the team involved in these studies, I am very grateful with your suggestions that were very helpful to improve our article. The corrections were incorporated in the revised manuscript. The following text has detailed point-to-point responses.

The present paper assessed the potential of agricultural practices in the control of Fusarium and mycotoxins contamination in maize grains. The findings of the paper are of interest especially to maize farmers and other stakeholders including consumers.

Major comments

The authors need to thoroughly work on the manuscript to correct language flaws.

Comments

Line 12, change “the mycotoxins” to “mycotoxin”

Done

Line 12, first time use of the abbreviation “DON”. Write the meaning

Done. “Deoxynivalenol” added.

Line 16-18, correct sentence

Done.

Line 20-23, correct sentence.

Done.

Line 31-32, correct sentence

Done.

Line 107-109, correct sentence

Done.

Line 113, “de”?

Corrected.

Line 184-186, Is this supposed to be part of the text?

Deleted.

Line 191-194, correct sentence

Done.

Line 217 – 222, the sentence is too long and confusing. Restructure and correct the sentence.

Done.

Line 218 -219, “attributed to differences in species composition”. How?

Corrected.

Line 253-255, what about the physical control methods?

Added.

Line 262-264, “However, some effectiveness…” Difficult to understand what the authors mean

Corrected.

Line 266-268, “However, F-BAC did not provide benefits…” This statement contradicts your earlier statement.

We added information to the results and discussion sections to clarify this question.

Line 293 – 294, correct sentence

Done.

Line 297, stablished?

Corrected.

Line 396, the manuscript for the first time mentioned that other mycotoxins including zearalenone and aflatoxins were analysed. This was not reported in the result and discussion sections. Even though the samples were negative of these toxins. The authors should report this in the result and discussion with possible explanation why this trend was observed in the study.

The following sentences were added to the first paragrapher of section 2.2.

“The analyses of mycotoxins included the quantification of fumonisins B1 (Fum B1) and B2 (Fum B2), deoxynivalenol (DON), toxin T2 (T2), zearalenone (ZEA) and aflatoxins (AFB1, AFB2, AFM1 and AFM2). However, only fumonisins B1 and B2 and DON were detected.”

“All the other mycotoxins values were below the detection limits of the used methods, being considered “not detected”.”

Reviewer 5 Report

Interesting work, making a significant contribution to the existing knowledge. Well written, but I have some minor comments:

Line 44-45: “maize is strongly devalued, also having a direct impact in the national economy” – I don’t understand what you mean

Line 59-60: “since these toxins result from exposure of the fungi to stressful conditions” – is it the only reason, what about the genetic background, are you able to show that the reduced production of mycotoxins was only due to less stress after using Nefusoil (no statistically significant difference)

Fig. 9: why there is a description on the axis “average”?, the values are shown as minimum and maximum

-          why was the surface disinfection of the grain, did it not get rid of fungi (including Fusarium) inhabiting the surface of the seeds in this way?

-          why incubation was carried out at 27°C (in the references it was 25°C)

Fig. 1, 2, 3, 4 – N Fusarium – what does it mean? Number?

Table 1 “variation between years”- what does it mean: variance (not in percent) or the coefficient of variation?

Line 184-186: “This section may be divided by subheadings. It should provide a concise and precise description of the experimental results, their interpretation, as well as the experimental

conclusions that can be drawn.” – what is this?

-          you write that the standards for individual mycotoxins were exceeded, or for all samples?, maybe it would be good to include these threshold values in the tables

-          are there exact standards for grain infestation by Fusarium and for the content of toxins that would disqualify grain as feed or seed, if so, it would be worth specifying.

Author Response

Dear reviewer

Many thanks for helping us with this Manuscript. On behalf of the team involved in these studies, I am very grateful with your suggestions that were very helpful to improve our article. The corrections were incorporated in the revised manuscript. The following text has detailed point-to-point responses.

Interesting work, making a significant contribution to the existing knowledge. Well written, but I have some minor comments:

Line 44-45: “maize is strongly devalued, also having a direct impact in the national economy” – I don’t understand what you mean

We already corrected that. We mean that the devaluation of maize due of its contamination with mycotoxins has a direct impact in the national economy, because maize has a high economic value.

Line 59-60: “since these toxins result from exposure of the fungi to stressful conditions” – is it the only reason, what about the genetic background, are you able to show that the reduced production of mycotoxins was only due to less stress after using Nefusoil (no statistically significant difference)

The study relied on the use of only one maize variety. Therefore, the genetic background was constant throughout the two years of the study. Indeed, we may state that the genetic background had no effect. Fungi synthetize mycotoxins as a response to stress, which can be caused by several biotic and abiotic factors. We verified that using Nefusoil, it was possible to enhance a significative reduction in the contamination with fumonisins in several samples (including reducing to zero FB2), so we hypothesized that Nefusoil had a positive effect in the Fusarium behaviour.

Fig. 9: why there is a description on the axis “average”?, the values are shown as minimum and maximum

The values do not show the minimum and maximum of temperature, but the average of the maximal and minimal temperatures registered in each month.

why was the surface disinfection of the grain, did it not get rid of fungi (including Fusarium) inhabiting the surface of the seeds in this way?

The surface disinfection of the grains was done to ensure that the Fusarium isolates were from infected grains and not from contaminated grains, excluding possible environmental contaminations of maize and an overestimation of Fusarium infected maize grains.

why incubation was carried out at 27°C (in the references it was 25°C)

Corrected.

Fig. 1, 2, 3, 4 – N Fusarium – what does it mean? Number?

Corrected. Changed to “Nº of Fusarium isolates”

Table 1 “variation between years”- what does it mean: variance (not in percent) or the coefficient of variation?

This corresponds to the difference in percentage between the years. For example, in 2019 were detected 231.5 µg/Kg dw of FB2 and in 2020 193,1 µg/Kg dw, which means a difference of 38.4 µg/Kg dw between the 2 years, i.e. 16.6% of the greatest value detected in these years. The legend was clarified as well as the first value .

Line 184-186: “This section may be divided by subheadings. It should provide a concise and precise description of the experimental results, their interpretation, as well as the experimental conclusions that can be drawn.” – what is this?

We are sorry. This instruction was kept by mistake. Deleted.

You write that the standards for individual mycotoxins were exceeded, or for all samples?, maybe it would be good to include these threshold values in the tables

We do not understand this statement because we reported the opposite:

“All the mycotoxins were at levels below the maximum limits established by the EU.”

Are there exact standards for grain infestation by Fusarium and for the content of toxins that would disqualify grain as feed or seed, if so, it would be worth specifying.

I suppose that the referee wanted to mean “threshold or limit to the concentration” rather “standard”. Standard is a solution or a mix of different mycotoxins intended to be used as calibrants of analytical instruments, methods validation or as positive controls.

There are no thresholds established for fungal infestation. The thresholds for the mycotoxins were established and published by the European Commission as referred in line 46-47 (revised document). Many third countries have also regulations for several mycotoxins, with the acceptable limits in food and feed.

Round 2

Reviewer 2 Report

Familiarization with the corrected version of the article by the authors showed the following:

The authors of the paper link the results obtained on mycotoxins with the use of fertilizers, the composition of which they indicated on lines 78-79: "F-BAC and Nefusoil, zinc-based and potassium phosphide-based, respectfully" (page 2). Firstly, the authors confuse in the article potassium phosphite (formula K2(PHO3) and potassium phosphide (formula K3P) (lines 69,78). There is no difference for the authors. Secondly, on page 11 (lines 347-348), the authors give a different versions of the composition of Nefusoil: “Nefusoil (EIBOL Ibérica S. L. Valencia, Spain), a fertilizer composed by nitrogen, phosphorus, and potassium (N:P2O5:K2O= 4:12:3)”, in which potassium oxide and phosphorus oxide appear instead of potassium phosphite or phosphide, and nitrogen appears in an unknown form – ammonium or nitrate – is not clear. Moreover, the authors cite reference No. 18, which refers to the use of a mixture of calcium oxide (CaO) and potassium phosphite to slow the growth of Fusarium fungi by 11%-53%. In this regard, a decrease in mycotoxins’ concentrations should also be expected. The action of CaO leads to the alkalinization of the soil, to the transition of humic acids into a soluble form and to become more bioavailable for use by microorganisms as substrates, and the authors of the article under discussion did not use calcium oxide in any way, but draw a parallel with the processes taking place.

Instead of the concentrations of fertilizers used, the authors give doses in liters per hectare (line 387). And how much fertilizer was in a liter?

Conclusion: it is not possible to use the information on fertilizers that the authors discuss in the context of mycotoxins’ decreasing concentrations.

I recommend the authors do not raise the issue of this connection (between Nefusoil and micotoxins) in the article, and do not emphasize the role of fertilizers, but simply provide actual statistics on the concentrations of mycotoxins determined by them in the samples under study. The main conclusion of the authors (line 331: "It appears to guarantee less stressful conditions to fungi, preventing their stimulation to produce fumonisins") is general, meaning that farmers need to apply fertilizers. They do that all over the world.

Reviewer 5 Report

I am satisfied. All remarks and comments were taken into consideration and the manuscript was revised.

Author Response

Thank you very much for your positive comments.